# The Influence of Dopaminergic Medication on Regularity and Determinism of Gait and Balance in Parkinson’s Disease: A Pilot Analysis

**DOI:** 10.3390/jcm13216485

**Published:** 2024-10-29

**Authors:** Craig D. Workman, T. Adam Thrasher

**Affiliations:** 1Department of Radiology, University of Iowa Health Care, 200 Hawkins Dr., Iowa City, IA 52242, USA; 2Department of Health and Human Performance, University of Houston, 3875 Holman Street, 104 Garrison Gym, Houston, TX 77204, USA; 3Center for Neuromotor and Biomechanics Research, 4733 Wheeler Ave, Houston, TX 77204, USA

**Keywords:** nonlinear, sample entropy, recurrence quantification analysis, determinism, Parkinson’s disease, gait, balance

## Abstract

**Background/Objectives**: Understanding how dual-tasking and Parkinson’s disease medication affect gait and balance regularity can provide valuable insights to patients, caregivers, and clinicians regarding frailty and fall risk. However, dual-task gait and balance studies in PD most often only employ linear measures to describe movement regularity. Some have used nonlinear techniques to analyze PD performances, but only in the on-medication state. Thus, it is unclear how the nonlinear aspects of gait or standing balance are affected by PD medication. This study aimed to assess how dopaminergic medication influenced the regularity and determinism of joint angle and anterior–posterior (AP) and medial–lateral (ML) center of pressure (COP) path time-series data while single- and dual-tasking in PD. **Methods**: Sixteen subjects with PD completed single- and dual-task gait and standing balance trials for 3 min off and on dopaminergic medication. Sample entropy and percent determinism were calculated for bilateral hip, knee, and shoulder joints, and the AP and ML COP path. **Results**: There were no relevant medication X task interactions for either the joint angles series or the balance series. Instead, the results supported independent effects of medication, dual-tasking, or standing with eyes closed. Balance task difficulty (i.e., eyes open vs. eyes closed) was detected by the nonlinear analyses, but the nonlinear measures yielded opposing results such that standing with eyes closed simultaneously yielded less regular and more deterministic signals. **Conclusions**: When juxtaposed with previous findings, these results suggest that medication-induced functional improvements in people with PD might be accompanied by a shift from lesser to greater signal consistency, and the effects of dual-tasking and standing with eyes closed were mixed. Future studies would benefit from including both linear and nonlinear measures to better describe gait and balance performance and signal complexity in people with PD.

## 1. Introduction

Parkinson’s disease (PD) affects approximately 1 million people in the US and is the second most prevalent neurodegenerative movement disorder [1]. It is diagnosed in men more often than women with a ratio as high as 4:1 [2]. The cardinal PD motor symptoms are rest tremor, rigidity, bradykinesia, postural instability, and gait impairment [3,4,5]. These symptoms are often treated pharmacologically with dopamine replacement, dopamine agonists, and catechol-O-methyltransferase inhibitors [3,4,5,6,7], albeit with limited effects [8]. Additionally, a decrease in automaticity, i.e., when a given motor task is performed without attentional control [9], is also common in PD [10]. Automaticity is assessed using dual-task (DT) paradigms, during which subjects perform a primary task (e.g., walking or standing) concurrent with a secondary task (e.g., phoneme monitoring). Any detriments to the performance of the primary and/or secondary tasks in dual-task paradigms can be interpreted as dual-task interference [11,12]. Dual-task interference in PD gait and balance has been well-documented [9,10,13,14,15,16] and the impact of dopaminergic medication on gait and balance in PD has been previously investigated [17,18,19].

Most studies in this area have almost exclusively employed traditional, linear analyses such as temporal-spatial measures of gait (e.g., velocity and stride length) and descriptive statistics (e.g., mean and SD) of the center of pressure (COP) while standing. Although these linear metrics and statistics are informative, they do not consider complex features of the entire physiological signal or provide information about the regularity of gait or balance performance. To determine some of these complex traits (e.g., movement regularity) nonlinear analyses, such as Sample Entropy (SampEn) and Recurrence Quantification Analysis (RQA), are required. 

SampEn (*m*, *r*, N) is a non-biased representation of Approximate Entropy [20], a group of statistics that computes the regularity of a signal [21], and results in more consistent outcomes than Approximate Entropy [22]. SampEn is dependent on the length of compared runs (*m*), a similarity threshold for points within the series (*r*), and the number of data points within the series (*N*). The regularity of the signal is estimated by calculating the probability that a time series of length N and dimension *m* is analogous to a series of length N and dimension *m* + 1 within the specified tolerance window *r* [20]. The output is a unitless number from 0 to 2, with 0 representing perfect regularity (e.g., a pure sine wave) and 2 corresponding to random noise [20,22]. Generally, SampEn values are elevated (i.e., less regular) in older adults [23,24], which has been interpreted as decreased adaptability and complexity in motor control [25,26]. Thus, assessments that include nonlinear measures might bolster dysfunctional postural control determinations in this population.

RQA considers the recurrent behavior of a dynamic signal using recurrence plots, which are graphical representations of recurrent patterns derived from a one-dimensional time series. RQA requires several inputs (i.e., embedding dimension, time delay, and threshold) to observe and interpret patterns to quantify the recurrent aspects of the recurrence plots [27]. Outputs for RQA include the percent recurrence (%REC) and percent determinism (%DET) of the signal. %REC is the percentage of data points that fall within a specified radius or threshold. %DET is the percentage of these recurrent points that form diagonal lines parallel to the central diagonal line in the recurrence plot and serves as an estimation of the signal’s regularity. Thus, %REC provides different information about the signal than SampEn, while %DET serves a complementary role in estimating the regularity of the signal from a deterministic, as opposed to a probabilistic (SampEn), perspective.

Nonlinear analysis of human movement signals to better understand Parkinson’s Disease is an emerging area of interest [28]. A few studies have used nonlinear techniques to describe complex features of PD gait [29,30,31]. Nonlinear analyses describing balance have also been performed, but only while subjects were in the on-medication state [32,33]. It is important to understand movement regularity in PD because it is associated with frailty and fall risk in older adults [34,35]. Furthermore, it is also worthwhile to appreciate how task complexity affects performance regularity because most activities of daily living are DT in nature. Additionally, many PD subjects experience periods of being off-medication, the so-called wearing-off effect [36], while performing these DT activities. Thus, a DT paradigm aids in understanding how medication affects movement regularity in real-life situations. Understanding how both dual-tasking and PD medication together affect movement regularity will provide useful information to clinicians and caregivers to ensure that PD subjects are operating within their capabilities.

This study was a secondary analysis of previous joint angle and balance data [37,38] and aimed to assess the influence of dopaminergic medication on the nonlinear characteristics of gait and balance while dual-tasking for long durations (3 min). It was hypothesized that medication would improve the nonlinear aspects of the gait signals such that on-medication (ON) DT conditions would be significantly more regular (i.e., smaller SampEn, larger %DET) than off-medication (OFF) single-task (ST) conditions. Additionally, it was hypothesized that medication would improve the nonlinear aspect of balance signals such that ON-DT conditions with the eyes closed (EC) would be significantly more regular (i.e., smaller SampEn, larger %DET) than OFF-ST eyes-open (EO) conditions. SampEn and RQA were chosen from the various other nonlinear tools because of their analogous and relatively simple interpretations.

## 2. Materials and Methods

### 2.1. Subjects

Sixteen subjects (female = 4; prevalence ratio of PD is 4:1 [2]) with mild to moderate PD (i.e., Hoehn and Yahr I–III [39]) were recruited from PD-specific activity groups in the greater Houston area. Inclusion criteria were: (1) a diagnosis of PD from a movement disorder specialist, (2) on an unchanged regimen of dopaminergic medication for ≥3 months to avoid transient medication-related motor performance differences, and (3) able to stand and walk unassisted for ≥3 min. Subjects were excluded if they (1) had injuries or surgeries that caused unusual gait or stances, (2) scored < 24 or <17, respectively, on the Montreal Cognitive Assessment (MoCA) [40] or telephone MoCA [41], (3) experienced freezing of gait, (4) had deep brain stimulation, or (5) a diagnosis of dementia or other neurodegenerative diseases. This study was approved by the University of Houston’s Institutional Review Board and all subjects provided written informed consent.

### 2.2. Equipment and Tasks

Kinematic data were collected using the Xsens MVN Biomech Awinda wireless system (Xsens Technologies B.V., Enschede, The Netherlands), which includes 17 inertial motion trackers (triaxial accelerometers, gyroscopes, and magnetometers) placed at the body segments shown in Figure 1. Movement data were collected at 60 Hz by each tracker and integrated into a full-body kinematic model by the Xsens software (version: MVN 2018). Bilateral hip, knee, and shoulder angles in the sagittal plane were extracted. Balance data were collected using the NeuroCom Balance Master force platform (NeuroCom International Inc., Clackamas, OR, USA) operating at 100 Hz. The path of the COP in anterior–posterior (AP) and medial–lateral (ML) directions was determined. PD motor symptoms were assessed using the motor section of the Movement Disorder Society Unified Parkinson’s Disease Rating Scale [42] (MDS-UPDRS III). 

The two primary tasks were (1) overground gait at a self-selected speed and (2) standing balance with eyes open and eyes closed. Each primary task was performed in a single trial for 3 min. The secondary task was a phoneme monitoring task, during which the subjects listened to audio text (i.e., an unfamiliar fairytale) through on-ear headphones and counted the number of times a pre-determined word occurred. The subjects were instructed to perform the counting mentally (i.e., not tally with fingers) while attending to the details of the story well enough to answer questions about the content of the story at the end of the trial. Phoneme monitoring was ideal for the desired long-duration collection time, because it has face validity with real-life situations, such as conversing while walking or standing [43]. A long-duration data collection time provided sufficient steady-state data for the nonlinear analyses and mirrors the real-life situations the subjects experience daily. Several auditory recordings of ~195 s were prepared so each condition had a novel story and phoneme to tally. No practice or familiarization trials were performed.

### 2.3. Procedures

Testing always commenced with the OFF state and all trials (OFF and ON) were completed in one session. To ensure the subjects were in a stable OFF state, they were instructed to undertake a minimum 12 h overnight medication withdrawal, as is common in PD studies [8,44]. First, the subjects’ anthropometric measurements were taken and input into the Xsens software kinematic model and they were outfitted with the Xsens sensors. This provided consistent conditions for the non-gait trials (i.e., UPDRS III, ST phoneme monitoring, and balance trials) during both OFF and ON states. This was necessary because once the sensors were donned, the subjects did not remove them.

After the sensors were placed, testing always commenced with the UPDRS III. The subjects then performed trials in blocks organized by gait and balance tasks. These blocks were randomly ordered. For the gait block, the subjects performed, in random order: ST phoneme monitoring while seated comfortably in a quiet room (if not completed in a previous balance block), ST gait, and DT gait, which was a combination of the walking task and phoneme monitoring. For the balance block, the following tasks were performed in random order: ST phoneme monitoring (if not completed in a previous gait block), ST standing eyes open (STEO), ST standing eyes closed (STEC), DT standing eyes open (DTEO), and DT standing eyes closed (DTEC). As before, the DT conditions were a combination of the ST standing and phoneme monitoring tasks. Each condition was performed once for 3 min. When needed, the subjects rested in a chair for ≥1 min between conditions or blocks. Before all phoneme monitoring conditions, the subjects were reminded to attend to the story well enough to answer the questions and report the phoneme tally at the end of the trial. No other explicit instructions for directing attention were provided.

Once all OFF trials were completed, the subjects took their medication as normally prescribed for their first/morning dose. ON testing commenced ~45–60 min later, with more time provided to achieve a subjective ON state as needed. During this transition time, subject demographic information (i.e., weight, time since diagnosis, and PD medication and dosages) were collected. ON testing again started with the UPDRS III and proceeded with the same condition instructions as OFF, but with a new set of randomized blocks and randomized conditions.

### 2.4. Data Processing

Segment-relative joint angles (i.e., 0° in the standing, neutral position) were exported using the Xsens software and COP information was exported using the NeuroCom software (version 3.1.0). All data were imported into MATLAB (R2023a, The MathWorks, Natick, MA, USA) for analysis, which was performed on unfiltered data [45,46]. Both joint angle and COP data were cropped to remove the first and last 30 s, yielding 2 min of steady-state performance for the analyses. The regularity characteristics of the bilateral hip, knee, and shoulder joint angle series and the COP series were determined using SampEn and RQA. The Cross Recurrence Plot (CRP) Toolbox [47] for MATLAB was used to perform the RQA, and percent recurrence (%REC) and percent determinism (%DET) of the signal were extracted. For SampEn, a script that follows the algorithm developed by Richman and Moorman [22] was obtained from the MATLAB File Exchange [48].

It must be noted that choosing input parameters for RQA (i.e., embedding dimension, time delay, and threshold) and SampEn (i.e., embedding dimension and tolerance/radius; *m* and *r*, respectively) can be challenging [27,49]. Furthermore, the choice of input parameters has a noticeable effect on the outcomes. The CRP Toolbox also includes codes to aid in calculating and selecting RQA input parameters based on the characteristics of the input series. The toolbox functions were used to estimate embedding dimension (using false nearest neighbors [49]), time delay (using mutual information [49,50]), and threshold (estimated as 10% of the maximum phase space diameter [49,50]) for each series. Additionally, Pellecchia and Shockley [27] recommended choosing a threshold such that %REC is less than 5%, to avoid artificially inflating %DET [27,49], but not too close to 0%, to avoid floor effects. Accordingly, %REC between 1% and 5% was required for this study. Among the RQA input parameters, the threshold most directly impacts %REC [49] and, by extension, %DET. Thus, when the toolbox code failed to provide a threshold that satisfied the %REC criteria, 10% of the mean phase space diameter was used to estimate the threshold [49]. If this threshold still resulted in an out-of-bounds %REC, it was manually adjusted in increments of 0.02 until %REC was within range. Lastly, a minimum length of four points was chosen to define a line segment [50] as a more conservative requirement for %DET calculation.

Parameter selection for SampEn is less variable than RQA but still lacks specific guidelines. Similar to RQA, the choice of the SampEn input parameters *m*, *r*, and N relevantly impact the outcome and partly depend on the physiological source of the time series (e.g., a cardiac/respiratory series [20,22] vs. spatiotemporal gait measures [51]). Still, to help address this problem, Montesinos, et al. [24] systematically evaluated different *m*, *r*, and N inputs on SampEn regularity approximations using COP time-series data from healthy young adults, older fallers, and older non-fallers. In addition to verifying that SampEn was more stable than Approximate Entropy, their analysis also indicated that input parameter selection significantly affects the entropy calculation, such that larger *m* and *r* resulted in smaller, more regular outputs [24]. Furthermore, SampEn with *m* = 4 or 5 and *r* = 0.25–0.35 (by 0.05 increments) was the only measure, linear or nonlinear, that distinguished the balance performances of their three study groups. Thus, the SampEn input parameters for the current analysis were N = 7200 (60 Hz × 120 s = 7200), *m* = 4, and *r* = 0.25 for the joint angles and N = 12,000 (100 Hz × 120 s = 12,000), *m* = 4, and *r* = 0.25 for the COP data. The smallest recommended *m* and *r* from Montesinos, et al. [24] were chosen to avoid floor effects.

### 2.5. Statistical Analysis

Because PD often asymmetrically impacts patients, e.g., unilaterally present (mild) or unilaterally more severe (moderate), joint angles were stratified into more-affected and less-affected sides. Although the subjects were asked to report their subjective ON state before proceeding with ON testing, this was objectively verified via paired *t*-test of the UPRDS III scores. For the joint angles analysis, a repeated-measures ANOVA was performed with medication (OFF vs. ON) and task (ST vs. DT) as repeated measures. Dependent variables included %DET and SampEn for the bilateral hip, knee, and shoulder joint angle series. For the COP analysis, a 3-factor repeated-measures ANOVA with medication (OFF vs. ON), task (ST vs. DT), and eye condition (EO vs. EC) as repeated measures was performed. Dependent variables were %DET and SampEn in the anterior–posterior (AP) and medial–lateral (ML) directions. The assumptions for a repeated-measures ANOVA were investigated and sufficiently met, thus no transformations or other adjustments were required. *Post hoc* comparisons were performed to determine pairwise differences of significant interactions and Cohen’s **d** was computed as a measure of effect size. Significance was accepted at *p* ≤ 0.05, uncorrected, and the analysis was performed using GraphPad Prism 10.2.2 (GraphPad Software, San Diego, CA, USA).

## 3. Results

All subjects completed the testing conditions according to study protocol and complete datasets were used for analyses. Table 1 displays the subjects’ demographic information, including the levodopa-equivalent dose [52] of their prescribed medications (including combinations of carbidopa/levodopa, dopamine agonists, and COMT inhibitors). The UPRDS III when ON was significantly lower than OFF (*p* < 0.001, **d** = 2.5), indicating the 12 h medication withdrawal induced an OFF state. Table 2 and Table 3 contain the mean ± SD of %REC and the computed parameters input into the RQA for the joint angle series and the COP series, respectively. Data reported in the text are mean ± SD.

### 3.1. Joint Angles During Gait

Figure 2 displays the SampEn results for the joint angle analysis. There was a significant medication X task interaction for SampEn at the less-affected knee (*p* = 0.050). Pairwise testing clarified that OFF-DT (0.125 ± 0.030) was smaller (i.e., more regular) than OFF-ST (0.134 ± 0.030; *p* = 0.016, **d** = 0.3), ON-ST (0.134 ± 0.026; *p* = 0.014, **d** = 0.4), and ON-DT (0.135 ± 0.027; *p* = 0.008, **d** = 0.4). There were also significant task effects for the more-affected hip (ST = 0.114 ± 0.026, DT = 0.106 ± 0.018, *p* = 0.022) and the more-affected shoulder (ST = 0.206 ± 0.081, DT = 0.229 ± 0.086, *p* = 0.034). There were no other significant SampEn effects (*p* > 0.076). 

Figure 3 displays the %DET results. The less-affected shoulder had significant medication (OFF = 89.3% ± 1.7%, ON = 92.8% ± 0.8%; *p* = 0.046) and task (ST = 91.9% ± 2.0%, DT = 90.1% ± 2.9%; *p* = 0.035) effects. There was also a task effect (ST = 86.5% ± 1.01%, DT = 83.7% ± 1.02%; *p* = 0.025) for the more-affected shoulder. There were no other significant %DET effects (*p* > 0.269).

### 3.2. COP While Standing

Figure 4 displays the SampEn and %DET results in both the AP and ML directions. For SampEn in the AP direction, there was an eyes condition effect (EO = 0.056 ± 0.003, EC = 0.063 ± 0.003; *p* = 0.038) and a medication X eyes condition interaction (*p* = 0.016). Pairwise testing revealed that OFF-EO (0.054 ± 0.033) was smaller (i.e., more regular) than OFF-EC (0.065 ± 0.034, *p* < 0.001, **d** = 0.3) and ON-EC (0.060 ± 0.033, *p* = 0.01, **d** = 0.2), and that ON-EO (0.058 ± 0.043) was smaller than OFF-EC (0.065 ± 0.034, *p* = 0.015, **d** = 0.2). SampEn in the ML direction only had a medication effect (OFF = 0.127 ± 0.013, ON = 0.068 ± 0.006; *p* = 0.031). There were no other significant effects for SampEn in either direction (*p* > 0.072). 

For %DET in the ML direction, there was a medication effect (OFF = 72.8% ± 3.7%, ON = 79.3% ± 2.1%; *p* = 0.007), an eyes condition effect (EO = 74.7% ± 5.06%, EC = 77.3% ± 4.06%; *p* = 0.045), and a task X eyes condition interaction (*p* = 0.050). Pairwise testing revealed that STEO (72.9% ± 10.1%) was smaller (i.e., less deterministic) than STEC (78.8% ± 8.14%; *p* = 0.019, **d** = 0.6). There were no other COP findings for %DET (*p* > 0.063). 

## 4. Discussion

The primary hypothesis for this study was that medication would improve the nonlinear aspects of gait signals, such that ON-DT conditions would be significantly more regular (smaller SampEn) and deterministic (larger %DET) than OFF-ST conditions, and of balance signals, such that ON-DTEC would be more regular and deterministic than OFF-STEO. Despite the significant medication X task interaction of SampEn at the less-affected knee, seemingly driven by a low OFF-DT value, these results do not support this hypothesis. Still, other interesting findings bear discussion.

There were medication effects for %DET of the less-affected shoulder during walking and for SampEn and %DET in the ML direction during standing, indicating increases in the regularity and determinism, respectively, of these signals while ON. This is interesting because linear measures of temporal-spatial, joint angle, and static posturography analyses of these same subjects indicated significant improvements from medication [37,38]. These studies, taken in combination with well-accepted motor improvements in people with PD after taking dopaminergic medication [53], might suggest that signal consistency increases in concert with medication-induced motor amelioration in people with PD. This is intriguing because a certain degree of irregularity is associated with an increase in adaptability [54,55], while decreased regularity is often indicative of impaired performance [24,34,35]. This suggests an inverted-U type of interaction between functional performance and time-series regularity and/or determinism, as proposed by Stergiou, et al. [56]. In combination with a recent report of lower joint angle SampEn in people with PD—while ON—compared to neurologically healthy controls [57], the present results imply that dopaminergic medication might shift people with PD closer to the optimum range of the inverted U. Of course, this supposition necessitates additional investigation and comparison with neurologically healthy controls.

The dual-tasking effects of the present study had mixed outcomes. For the SampEn analysis of joint angles, dual-tasking resulted in greater signal regularity of the less-affected knee (see the pairwise results of the interaction effect) and the more-affected hip, while it had the opposite effect at the more-affected shoulder. The latter result seems more intuitive and agrees with a previous report of increased SampEn while dual-tasking in people with PD [58]. However, the former results counter this intuition but might suggest that dual-tasking yields a more rigid, less adaptive system in people with PD, which would result in lower SampEn values [59]. Considering that dual-tasking decreased cadence, stride length, and some joint angles in these same subjects [38], a more rigid, less adaptive system while dual-tasking is more likely and aligns with the inverted-U concept discussed above [56].

Comparably interesting findings were also present in the balance analysis. There were main and interaction effects involving differences in eye conditions, with EO yielding more regular results than EC, despite increases in task demands (i.e., ST vs. DT). These might be expected and imply that greater balance difficulty, via closing the eyes, resulted in decreased time-series predictability, which agrees with previous understandings of balance regularity and stability [33]. Furthermore, the lack of significant findings between EC conditions indicates that standing with EC alone might be sufficient to influence balance, such that executing a secondary task and/or taking PD medication were inadequate to further challenge or recover balance performance. Contrary to SampEn, the %DET calculation for the ML direction yielded the opposite result and EC conditions were more deterministic than EO. Interestingly, Schmit, et al. [33] reported greater determinism in PD patients compared to controls and suggested that this was a result of a loss of balance complexity. They further argued that abnormal neuromuscular activity during balance control, in the form of difficulty/slowness in terminating postural control movements, could increase the signal-to-noise ratio of the neuromuscular system and, thereby, increase %DET without improving balance performance *per se* [33]. However, future investigations would benefit from including neuromuscular data (e.g., electromyography) with %DET to better substantiate this hypothesis.

For many condition groupings, determinism and regularity values were highly deterministic and regular. This might indicate either floor/ceiling effects of the measures, within the confines of the input parameters, or that both gait and balance control in PD subjects are highly predictable, despite medication state, task complexity, and eye condition. The low to medium ranges of %DET in the balance analysis and how the present data fit within the broader research landscape suggest the latter supposition is more likely. Using linear variability measures (e.g., SD, coefficient of variation [CV]), ‘good gait’ and ‘good balance’ are often interpreted as less variability equating to better performance [13,60]. Similarly, disease progression in PD is associated with increased gait variability [61], which has been related to falls [62]. However, others have reported increased temporal predictability using nonlinear metrics in PD [33,63]. Thus, the similarity of the present nonlinear data across medication states, tasks, and conditions challenges these traditional, nonlinear interpretations. As above, this is especially relevant considering that these same subjects experienced significant gait and balance changes from these medication states, task complexities, and balance conditions [37,38]. Thus, novel interpretations of gait and balance performance exclusive to nonlinear measures, and separate from linear analyses, especially with PD subjects, might be required. However, the application of nonlinear tools in PD movement, especially PD gait, is still burgeoning, and more research comparing nonlinear metrics of PD subjects with neurologically healthy age-matched and young controls is necessary.

The modest number of subjects in the analysis is a potential limitation of the current study and including more subjects might reveal more subtle differences between the conditions. Another limitation was the lack of a neurologically healthy control group. However, given that the primary thrust of this study was to investigate the effects of dopaminergic medication on dual-tasking in PD, and given the well-established dual-tasking differences between people with PD and neurologically healthy controls [62,64], the inclusion of a comparator group was not part of the original research plan. Another potential limitation is that the data and interpretations were derived from a single trial and might not reflect an individual’s “true” performance. The longer trial durations and analysis of steady-state data (i.e., the middle two minutes) were selected to provide sufficient data points for the nonlinear analysis and to increase the face validity and applicability to real-life situations. Still, day-to-day and diurnal symptom variations and responses to medication might have influenced the results—but this limitation is common to all single-session studies in people with PD and also represents a source of variability for multiple-session research. Lastly, this study only included people with mild–moderate PD and how or if these findings apply to prodromal, *de novo*, or severe PD remains to be investigated. If these nonlinear metrics can help discriminate disease severities and/or medication-related responses (when applicable), such analyses might bolster diagnosis certainty. Additionally, once appropriate input parameters are selected (see Table 2 and Table 3), these metrics could readily be applied in clinics that employ force plate measurements. Future studies are encouraged to include both linear and nonlinear measures in their analyses to better capture both temporal-spatial, kinematic, and signal consistency metrics. Illuminating the interplay between these metrics might inform targeted therapeutic and motor control research in PD. Furthermore, investigations that include nonlinear analyses in patient populations similar to PD, e.g., atypical parkinsonism and Progressive Supranuclear Palsy-Parkinsonism Predominant [65,66], might help contrast the motor differences between these groups and bolster diagnosis certainty.

## 5. Conclusions

Dopaminergic medication did not improve dual-task predictability of joint angle or balance signals but independently altered the regularity and determinism in these time series. When juxtaposed with improvements in motor function, these results suggest that medication-induced functional improvements in people with PD might be accompanied by a shift from lesser to greater predictability. Balance task difficulty (i.e., eyes open vs. eyes closed) was detected by the nonlinear analyses, but the nonlinear measures returned opposing results such that standing with eyes closed simultaneously yielded less regular and more deterministic signals. Future studies would benefit from including both linear and nonlinear measures to better describe gait and balance performance and signal complexity in people with PD.

## Figures and Tables

**Figure 1 jcm-13-06485-f001:**
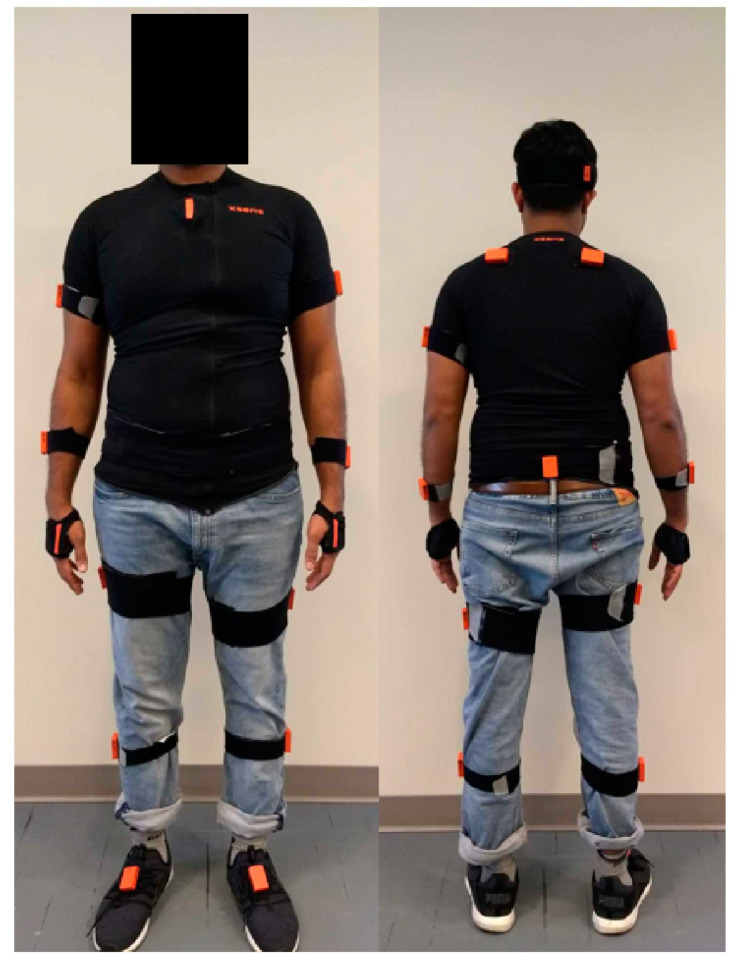
Xsens sensor locations (17 total) on the head, sternum, posterior pelvis (i.e., L5/Sacrum), and bilaterally on the shoulders, upper arms, forearms, hands, thighs, lower legs, and feet. Sensors are shown on top of straps for visualization.

**Figure 2 jcm-13-06485-f002:**
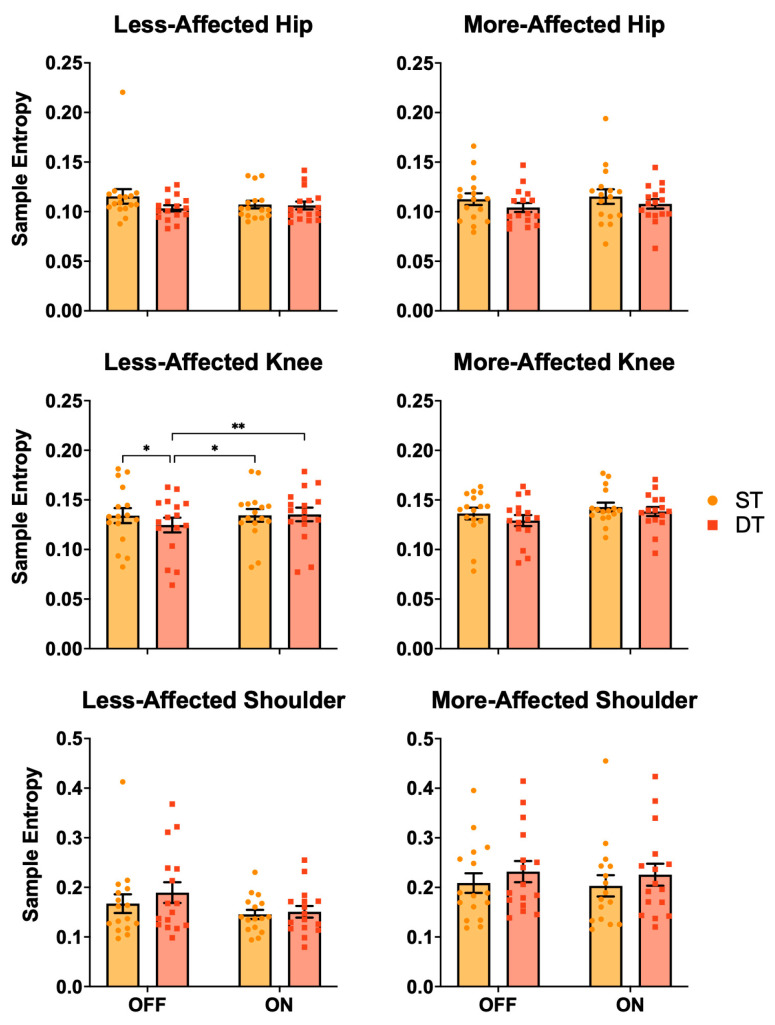
Bar graphs depicting the mean ± SEM sample entropy (SampEn) on the less- and more-PD-affected joints. There was a medication X task interaction for SampEn at the less-affected knee, with OFF-DT smaller (i.e., more regular) than OFF-ST, ON-ST, and ON-DT. There were also task effects for the more-affected hip (ST > DT) and the more-affected shoulder (ST < DT). * = *p* < 0.05; ** = *p* < 0.01; PD = Parkinson’s disease; ST = single task; DT = dual task; OFF = off-medication state; ON = on-medication state.

**Figure 3 jcm-13-06485-f003:**
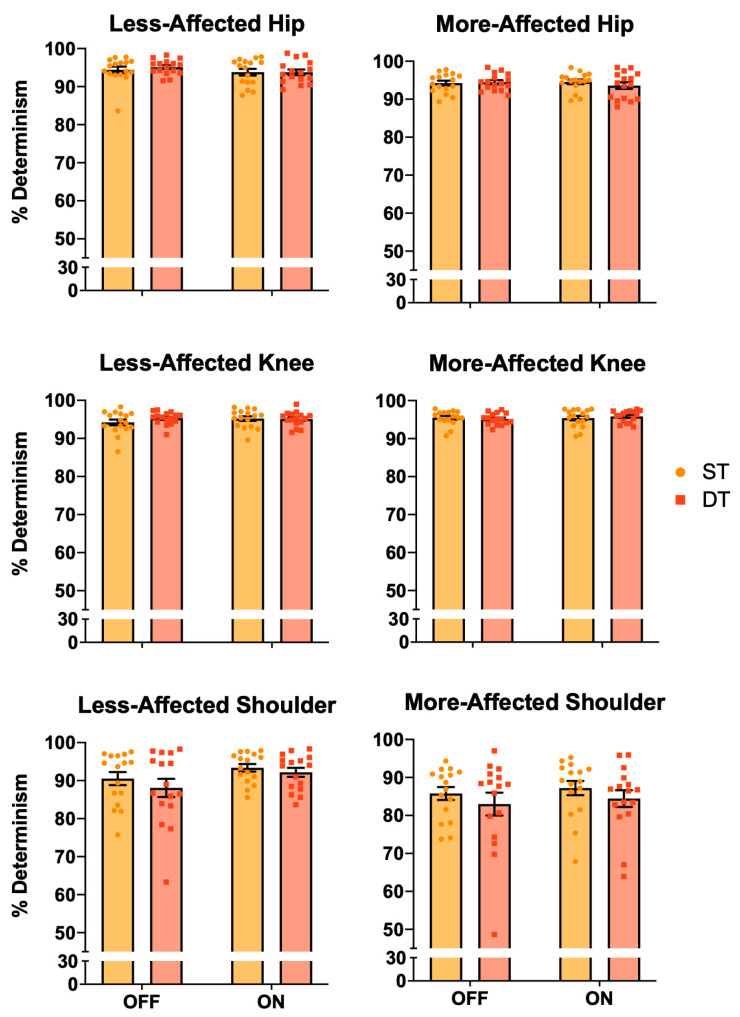
Bar graphs depicting the mean ± SEM percent determinism (%DET) on the less- and more-PD-affected joints. The less-affected shoulder had medication (OFF < ON) and task (ST > DT) effects. There was also a task effect (ST > DT) for the more-affected shoulder. PD = Parkinson’s disease; ST = single task; DT = dual task; OFF = off-medication state; ON = on-medication state.

**Figure 4 jcm-13-06485-f004:**
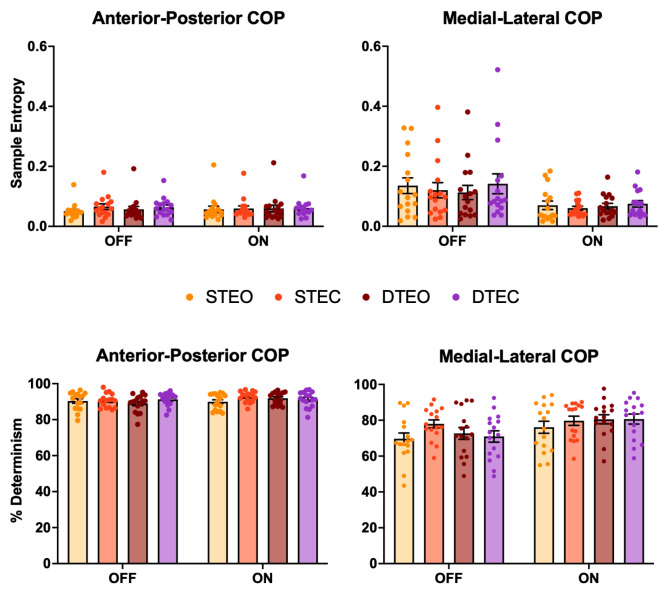
Bar graphs depicting the mean ± SEM sample entropy (SampEn; top row) percent determinism (%DET; bottom row) for the anterior–posterior (AP; left) and medial–lateral (ML; right) center of pressure (COP) directions. For SampEn in the AP direction, there was an eyes condition effect (EO < EC) and a medication X eyes condition interaction, with OFF-EO smaller (i.e., more regular) than OFF-EC and ON-EC, and ON-EO smaller than OFF-EC. SampEn in the ML direction only had a medication effect (OFF > ON). For %DET in the ML direction, there was a medication effect (OFF < ON), an eyes condition effect (EO < EC), and a task X eyes condition interaction, with STEO smaller (i.e., less deterministic) than STEC. STEO = single-task, eyes open; STEC = single-task, eyes closed; DTEO = dual-task, eyes open; DTEC = dual-task, eyes closed; OFF = off-medication state; ON = on-medication state.

**Table 1 jcm-13-06485-t001:** Subject demographic information. Data are mean ± SD, where appropriate.

Demographics	
Sex	12 male, 4 female
Age	67.1 ± 7.5 years
Height	171.2 ± 9.5 cm
Weight	80.9 ± 14.3 kg
UPDRS III (OFF)	44.4 ± 13.3
UPDRS III (ON)	24.4 ± 8.2 *
More-affected side	Right = 9, Left = 7
Time since diagnosis	6.7 ± 5.8 years
Levodopa-Equivalent Dose	669.5 ± 230.6

Note: UPDRS III = part III of the MSD-UPDRS, OFF = off-medication state, ON = on-medication state. * Significantly smaller than OFF.

**Table 2 jcm-13-06485-t002:** The %REC and calculated parameters input into the RQA for the joint angle series. Data are mean ± SD.

RQA Parameter	Condition							
	OFF-ST	OFF-DT	ON-ST	ON-DT	OFF-ST	OFF-DT	ON-ST	ON-DT
	More-Affected Side				Less-Affected Side			
Hip								
%REC	3.28 ± 1.10	2.28 ± 1.08	2.21 ± 0.69	2.13 ± 0.84	2.51 ± 1.03	2.67 ± 1.38	2.76 ± 1.27	2.91 ± 1.35
Dimension	6.19 ± 1.52	6.25 ± 1.06	6.25 ± 1.29	6.31 ± 1.30	6.75 ± 1.61	6.63 ± 1.09	6.31 ± 1.20	6.69 ± 1.78
Time Delay	2.00 ± 0.00	2.00 ± 0.00	2.00 ± 0.00	2.00 ± 0.00	2.00 ± 0.00	2.00 ± 0.00	2.00 ± 0.00	2.00 ± 0.00
Threshold	0.29 ± 0.12	0.24 ± 0.07	0.22 ± 0.09	0.22 ± 0.08	0.27 ± 0.13	0.27 ± 0.10	0.25 ± 0.11	0.28 ± 0.12
Knee								
%REC	4.06 ± 0.82	3.60 ± 1.08	4.05 ± 1.12	3.88 ± 1.23	3.85 ± 1.10	3.92 ± 0.97	4.42 ± 0.59	3.91 ± 1.14
Dimension	5.50 ± 0.73	5.81 ± 1.33	5.44 ± 0.89	5.81 ± 1.11	5.50 ± 1.59	5.50 ± 0.82	5.38 ± 0.62	5.63 ± 1.26
Time Delay	2.00 ± 0.00	2.00 ± 0.00	2.00 ± 0.00	2.00 ± 0.00	2.00 ± 0.00	2.00 ± 0.00	2.00 ± 0.00	2.00 ± 0.00
Threshold	0.26 ± 0.03	0.27 ± 0.03	0.25 ± 0.02	0.27 ± 0.03	0.25 ± 0.05	0.27 ± 0.04	0.25 ± 0.04	0.27 ± 0.04
Shoulder								
%REC	1.24 ± 0.43	1.57 ± 1.05	1.32 ± 0.81	1.48 ± 0.90	2.23 ± 1.20	2.07 ± 0.85	2.34 ± 1.44	2.10 ± 1.38
Dimension	6.19 ± 1.60	6.19 ± 1.56	6.44 ± 1.31	6.38 ± 1.82	5.63 ± 0.96	6.00 ± 1.59	5.88 ± 1.36	5.88 ± 1.36
Time Delay	2.00 ± 0.00	2.00 ± 0.00	1.88 ± 0.34	1.94 ± 0.25	1.94 ± 0.25	2.00 ± 0.00	1.94 ± 0.25	2.00 ± 0.00
Threshold	0.33 ± 0.18	0.37 ± 0.24	0.36 ± 0.16	0.41 ± 0.26	0.31 ± 0.10	0.30 ± 0.07	0.35 ± 0.14	0.29 ± 0.09

Note: RQA = Recurrence Quantification Analysis, %REC = percent recurrence, OFF = off-medication state, ON = on-medication state, ST = single-task, DT = dual-task.

**Table 3 jcm-13-06485-t003:** The %REC and calculated parameters input into the RQA for the AP and ML COP series. Data are mean ± SD.

RQA Parameter	Condition							
	STEO	STEC	DTEO	DTEC	STEO	STEC	DTEO	DTEC
	AP Direction				ML Direction			
OFF								
%REC	2.09 ± 0.73	1.78 ± 0.61	1.83 ± 0.55	2.07 ± 0.79	1.71 ± 0.66	2.16 ± 0.55	1.85 ± 0.64	1.96 ± 0.80
Dimension	7.19 ± 1.42	6.44 ± 0.89	6.81 ± 0.83	6.96 ± 0.95	7.69 ± 1.25	7.63 ± 0.89	7.75 ± 1.13	7.50 ± 1.03
Time Delay	2.00 ± 0.00	2.00 ± 0.00	2.00 ± 0.00	2.00 ± 0.00	1.94 ± 0.25	1.94 ± 0.25	1.88 ± 0.34	1.94 ± 0.25
Threshold	0.20 ± 0.06	0.19 ± 0.06	0.20 ± 0.06	0.21 ± 0.05	0.31 ± 0.14	0.32 ± 0.14	0.28 ± 0.11	0.31 ± 0.13
ON								
%REC	1.74 ± 0.56	1.99 ± 0.64	2.20 ± 0.61	1.99 ± 0.59	1.76 ± 0.50	2.00 ± 0.55	2.12 ± 0.68	2.21 ± 0.66
Dimension	7.31 ± 1.20	6.81 ± 0.83	6.94 ± 1.34	6.75 ± 0.68	8.06 ± 1.57	7.50 ± 1.26	7.94 ± 1.24	8.00 ± 1.51
Time Delay	2.00 ± 0.00	1.94 ± 0.25	2.00 ± 0.00	2.00 ± 0.00	2.00 ± 0.00	2.00 ± 0.00	1.94 ± 0.25	1.94 ± 0.25
Threshold	0.20 ± 0.06	0.20 ± 0.05	0.22 ± 0.08	0.21 ± 0.06	0.22 ± 0.08	0.23 ± 0.06	0.25 ± 0.10	0.26 ± 0.07

Note: RQA = Recurrence Quantification Analysis, %REC = percent recurrence, AP = anterior–posterior direction, ML = medial–lateral direction, STEO = single-task eyes open, STEC = single-task eyes closed, DTEO = dual-task eyes open, DTEC = dual-task eyes closed, OFF = off-medication state, ON = on-medication state.

## Data Availability

Data that support this article will be made available upon request to the corresponding author.

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
