# Peer review of "The Influence of Dopaminergic Medication on Regularity and Determinism of Gait and Balance in Parkinson’s Disease: A Pilot Analysis"

_jcm, 2024, doi:10.3390/jcm13216485_

Round 1

Reviewer 1 Report

Comments and Suggestions for Authors

Workman and Thrasher elaborate on the influence of dopaminergic medication in the context of regularity and determinism of gait and balance in Parkinson's Disease (PD). I have the following comments regarding the work:

1. In the introduction authors should elaborate on the overlapping of PD and other parkinsonisms e.g. atypical parkinsonisms in which the issues concerning gait and balance are crucial. Moreover these clinical entities overlap in the context of clinical stages in early stages with PD e.g. Progressive Supranuclear Palsy-Parkinsonism Predominant (PSP-P). 

Ref.

Alster P, Madetko-Alster N. Significance of dysautonomia in Parkinson's Disease and atypical parkinsonisms. Neurol Neurochir Pol. 2024;58(2):147-149. doi: 10.5603/pjnns.98678. Epub 2024 Mar 19. PMID: 38501557.

"Parkinson's disease" on the way to progressive supranuclear palsy: a review on PSP-parkinsonism. Neurol Sci. 2021 Dec;42(12):4927-4936. doi: 10.1007/s10072-021-05601-8. Epub 2021 Sep 17. PMID: 34532773

2. Due to the partial resemblance authors could discuss whether in their opinion could hypothetically be feasible in PSP-P.

3. The issue of dopaminergic treatment should be discussed in the context of possible methods - levodopa, dopaminergic agonists etc.

4. The number of patients is very low. Moreover the proportion of males and females is unbalanced. As authors indicated the work lacks neurologically healthy control group. In this context perhaps this work should be considered more as a preliminary study rather than full-length article. Moreover the limitations of the study should be more stressed. 

5. In the context of the statement emphasized in the conclusion "Dopaminergic medication did not improve dual-task predictability of joint angle or  balance signals but independently altered the regularity and determinism in these time series. ",  authors should provide a discussion on clinical feasibility and future perspectives.

Author Response

Reviewer #1: Workman and Thrasher elaborate on the influence of dopaminergic medication in the context of regularity and determinism of gait and balance in Parkinson's Disease (PD). I have the following comments regarding the work:

  1. In the introduction authors should elaborate on the overlapping of PD and other parkinsonisms e.g. atypical parkinsonisms in which the issues concerning gait and balance are crucial. Moreover these clinical entities overlap in the context of clinical stages in early stages with PD e.g. Progressive Supranuclear Palsy-Parkinsonism Predominant (PSP-P).

Ref. Alster P, Madetko-Alster N. Significance of dysautonomia in Parkinson's Disease and atypical parkinsonisms. Neurol Neurochir Pol. 2024;58(2):147-149. doi: 10.5603/pjnns.98678. Epub 2024 Mar 19. PMID: 38501557.

 "Parkinson's disease" on the way to progressive supranuclear palsy: a review on PSP-parkinsonism. Neurol Sci. 2021 Dec;42(12):4927-4936. doi: 10.1007/s10072-021-05601-8. Epub 2021 Sep 17. PMID: 34532773

 RESPONSE: Although we find this commonality and potential link between PD and PSP-P fascinating, the focus of our manuscript is on people with a diagnosis of PD at the time of participation and if/how their dopaminergic medication affects the nonlinear characteristics of their gait and balance signals. As such, we feel an in-depth discussion of other diseases would distract from our primary purpose. Still, we find value in highlighting potential avenues for future work, when appropriate. Accordingly, we added text to the end of the Discussion (lines 402-405) suggesting this work be performed.

  1. Due to the partial resemblance authors could discuss whether in their opinion could hypothetically be feasible in PSP-P.

RESPONSE: See our response to the previous comment.

  1. The issue of dopaminergic treatment should be discussed in the context of possible methods - levodopa, dopaminergic agonists etc.

 RESPONSE: We already mention, albeit very briefly, some of the dopaminergic medications employed as pharmacological treatments and provide several citations for them and their effectiveness. Additionally, because our study dealt with the effect of any prescribed combination of medications on the nonlinear characteristics of gait and balance time-series, we felt a deeper exploration of the purpose and action of the different types of medications would distract from the message and was, therefore, not included. Instead, we have included a brief description of the relevant efficacy, and insufficiency, of PD medications in general, as this fits better with our primary purpose (see lines 41-43). A study that would justify a discussion you mention would be interesting, but also difficult to perform with a homogenous subject sample. We also added the existence of the levodopa equivalent dose calculation, with a citation, and mentioned the types of medications the subjects were taking (lines 246-247).

  1. The number of patients is very low. Moreover the proportion of males and females is unbalanced. As authors indicated the work lacks neurologically healthy control group. In this context perhaps this work should be considered more as a preliminary study rather than full-length article. Moreover the limitations of the study should be more stressed.

 RESPONSE: The ratio of males to females is on par with diagnostic incidence (4:1; see the newly included epidemiology reference in the Introduction – lines 38-39 – and Methods - line 110) and, as mentioned in the limitations, a neurologically healthy control group (who would not be taking PD medication – a key aspect of the study) was not a priority and already mentioned in as a limitation. Still, the modest number of subjects is a potential limitation that was not previously mentioned and has been added to the final paragraph of the Discussion (liens 378-380) and we have edited the title to include “A Pilot-Analysis” (lines 3-4).

  1. In the context of the statement emphasized in the conclusion "Dopaminergic medication did not improve dual-task predictability of joint angle or balance signals but independently altered the regularity and determinism in these time series. “, authors should provide a discussion on clinical feasibility and future perspectives.

 RESPONSE: We appreciate this comment and the response it has engendered from us (lines 393-398). We have updated the final paragraph of the Discussion to include the potential clinical relevance of our findings; in so doing we have highlighted an additional study limitation (i.e., the limited disease severity of our sample). 

Reviewer 2 Report

Comments and Suggestions for Authors

The study employed advanced nonlinear measures like Sample Entropy and Recurrence Quantification Analysis to assess gait and balance in Parkinson's disease, which provides fresh insights into understanding movement patterns beyond traditional linear metrics. I have several major and minor comments for the authors to address. 

Specific comments:

1.      The sentence "An appreciation for how dual-tasking and Parkinsons disease medication affect gait and balance regularity can inform patients, caregivers, and clinicians about frailty and fall risk." can be revised to "Understanding how dual-tasking and Parkinsons disease medication affect gait and balance regularity can provide valuable insights to patients, caregivers, and clinicians regarding frailty and fall risk." to clarity the manuscript.

2.      Given that the inclusion criteria require subjects to have been on medication for more than 3 months, the effects of the medication may be long-term. It is necessary to discuss whether the impact of a single medication withdrawal is reasonable in the Introduction.

3.      The logic in the Introduction is somewhat unclear. The Introduction should clearly address several main points: 1) the advantages of nonlinear analysis and the two key metrics (SampEn and RQA), 2) the differences between dual-task and single-task performance in Parkinson's disease patients, and 3) the effects of dopaminergic medication on gait and balance in Parkinson's disease patients. Currently, the nonlinear content is somewhat scattered, for example, in the 2nd, 3rd, and 4th paragraphs, as well as the first sentence of the 5th paragraph and the end of the 6th paragraph. These nonlinear-related discussions should be consolidated and refined. The Introduction should focus on introducing SampEn and RQA as the two primary nonlinear metrics, while the details of how they are calculated should be placed in the Methods section.

4.      Additionally, also in the Introduction, the first sentence of the 5th paragraph introduces nonlinear analysis, but the following content shifts to discussing medication and dual-tasking. It is recommended that this paragraph also be more focused on its theme, such as introducing the dual-task paradigm more clearly.

5.      The hypotheses should be more specific, as there are two distinct parts: gait and balance. For the gait test, there are two variables: medication (on or off) and the dual-task versus single-task mode. For the balance test, there are three variables: medication, task complexity (dual-task versus single-task), and eye condition (eyes open or closed). The hypotheses should be at least divided into two, addressing gait and balance separately.

6.      Although MATLAB's CRP toolbox is well-established for calculating SampEn and RQA, the specific algorithms used should be detailed in the Methods section to allow readers to learn from and replicate the process.

7.      Regarding the statistical methods, since the authors stated, The assumptions for a repeated measures ANOVA were investigated and sufficiently met, and no adjustments were required, does this mean that the p-values not adjusted for multiple comparisons in the post-hoc tests? The p-values should be adjusted for multiple comparisons.

8.      Is there no statistical analysis for the data presented in Tables 2 and 3?

9.      In Figures 2 and 3, all statistically significant results should be indicated, as currently only at the less-affected knee are marked. Additionally, the x-axis of each subplot should be clearly labeled with "OFF" and "ON".

10.   The limitations section should include a discussion of the immediate and long-term effects of the medication.

Author Response

Reviewer #2: The study employed advanced nonlinear measures like Sample Entropy and Recurrence Quantification Analysis to assess gait and balance in Parkinson's disease, which provides fresh insights into understanding movement patterns beyond traditional linear metrics. I have several major and minor comments for the authors to address.

 Specific comments:

  1. The sentence "An appreciation for how dual-tasking and Parkinson’s disease medication affect gait and balance regularity can inform patients, caregivers, and clinicians about frailty and fall risk." can be revised to "Understanding how dual-tasking and Parkinson’s disease medication affect gait and balance regularity can provide valuable insights to patients, caregivers, and clinicians regarding frailty and fall risk." to clarity the manuscript.

RESPONSE: We appreciate the clarification your revision provides and have made the requested change (lines 12-14).

  1. Given that the inclusion criteria require subjects to have been on medication for more than 3 months, the effects of the medication may be long-term. It is necessary to discuss whether the impact of a single medication withdrawal is reasonable in the Introduction.

RESPONSE: Because a change in medication is associated with poorly managed symptoms, this criterion was to ensure the subjects were on an established-as-effective medication regimen and avoid transient differences in motor performance as the result of changing medication dosing (see the newly added clarification of this on line 114). Furthermore, the 12-hour medication withdrawal is common practice in studies of people with PD where medication is a variable of interest; citations justifying this were added to the Methods (lines 152-153). We also added text to the Statistical Analysis (lines 229-231) and Results (lines 247-249) highlighting that UPDRS III scores were significantly different between OFF and ON testing (i.e., the 12-hour withdrawal successfully resulted in an OFF state).

  1. The logic in the Introduction is somewhat unclear. The Introduction should clearly address several main points: 1) the advantages of nonlinear analysis and the two key metrics (SampEn and RQA), 2) the differences between dual-task and single-task performance in Parkinson's disease patients, and 3) the effects of dopaminergic medication on gait and balance in Parkinson's disease patients. Currently, the nonlinear content is somewhat scattered, for example, in the 2nd, 3rd, and 4th paragraphs, as well as the first sentence of the 5th paragraph and the end of the 6th paragraph. These nonlinear-related discussions should be consolidated and refined. The Introduction should focus on introducing SampEn and RQA as the two primary nonlinear metrics, while the details of how they are calculated should be placed in the Methods section.

RESPONSE: The emphasis on the nonlinear aspects of the study is intentional. Specifically, the other points you mention (especially 2 and 3) are well-established in the literature and were, therefore, only briefly highlighted in the Introduction. We feel that additional details in this manuscript would distract from the stated purpose of the analysis. Accordingly, the logical flow of the Introduction is: 1) A brief overview of PD, medication treatment efficacy, and automaticity and dual-task paradigms (addressing points 2 and 3 of your comment); 2) introduce the concept of nonlinear analyses and contrast them with linear analyses (point 1 of your comment); 3) preliminary details about SampEn, including a textual, non-methodological description of the input parameters and interpretation (details on choosing input parameters are saved for the Methods); 4) preliminary details about RQA with the same goal as 3; 5) how nonlinear analyses fit in the gait/balance-in-PD and medication-in-PD research landscape (this is very niche and our studies are among the very few that are specifically looking at this combination of effects; i.e., medication + dual-tasking gait/balance, and now adding nonlinear metrics with this analysis); and 6) state the purpose and context of the study, the aim of this follow-up analysis, and our analysis hypothesis and variable selection justification. To your point, we also had some internal discussions about what aspects of SampEn and RQA were appropriate for the Introduction and what should be saved for the Methods. In the end, we felt the current balance was the most ideal for providing sufficient information to understand the purpose and interpretation of these metrics in the Introduction and saving the details about choosing the input parameters (and how that affects the outcomes) for the Methods.

  1. Additionally, also in the Introduction, the first sentence of the 5th paragraph introduces nonlinear analysis, but the following content shifts to discussing medication and dual-tasking. It is recommended that this paragraph also be more focused on its theme, such as introducing the dual-task paradigm more clearly.

RESPONSE: As above, this introductory sentence about nonlinear analyses is intentional as it helps set the stage for describing the research landscape in which we are operating.

  1. The hypotheses should be more specific, as there are two distinct parts: gait and balance. For the gait test, there are two variables: medication (on or off) and the dual-task versus single-task mode. For the balance test, there are three variables: medication, task complexity (dual-task versus single-task), and eye condition (eyes open or closed). The hypotheses should be at least divided into two, addressing gait and balance separately.

RESPONSE: We agree that more specific hypotheses for each motor performance are justified. See the added text in lines 102-106 and lines 298-300 (beginning of the Discussion).

  1. Although MATLAB's CRP toolbox is well-established for calculating SampEn and RQA, the specific algorithms used should be detailed in the Methods section to allow readers to learn from and replicate the process.

RESPONSE: First, as a point of clarification, the CRP Toolbox was only used for the RQA, while a different script was used for SampEn. This is now more clearly described in lines 189-191. Second, given the clinical target of readers of the Journal of Clinical Medicine, we have avoided repeating computation formulae. Instead, we have provided multiple citations and stated the programs and scripts that underpin our analyses. Thus, we feel we have provided sufficient information for interested and intrepid readers to replicate our analysis methods while maintaining access of our results to a broader set of readers.

  1. Regarding the statistical methods, since the authors stated, “The assumptions for a repeated measures ANOVA were investigated and sufficiently met, and no adjustments were required,” does this mean that the p-values not adjusted for multiple comparisons in the post-hoc tests? The p-values should be adjusted for multiple comparisons.

RESPONSE: The “adjustments” mentioned here are specific to those one might make if some of the assumptions are not met. For example, if normality is not assumed, one might log-transform the data or if sphericity is violated, one might use a Greehouse-Geisser degrees of freedom correction. This statement is meant to clarify those types of adjustments/corrections. We edited the text for clarity (line 238). Regarding p-value correction, given the preliminary/pilot nature of the analysis, we felt it was better to report uncorrected results to help orient future studies. We clarified that the p-value was not corrected for multiple tests on line 241.

  1. Is there no statistical analysis for the data presented in Tables 2 and 3?

RESPONSE: No, we did not investigate the differences in the RQA input parameters. Statistical tests of %REC have been performed by other researchers, but our limitation of %REC between 1% and 5% might bias such an analysis. Furthermore, statistically testing differences in dimension, time delay, and threshold between condition combinations is not a common practice. We provide them for full transparency and context for our RQA outputs and to provide potential parameter ranges for interested researchers (see added text in lines 396-398).

  1. In Figures 2 and 3, all statistically significant results should be indicated, as currently only at the less-affected knee are marked. Additionally, the x-axis of each subplot should be clearly labeled with "OFF" and "ON".

RESPONSE: Only pairwise effects of significant interactions are indicated in the figures because main effects of task, medication, and eyes condition would collapse individual data columns together and necessitate additional figures. Moreover, the interaction effects (2-way for gait and 3-way for balance) are the most relevant for our hypothesized differences. Regarding x-axis labels, this might partially be a matter of personal preference, but we feel that including repeated x- and y-axis labels in subplots is too busy and often confuses the subplot titles and neighboring data. Thus, only the plots on the left have y-axis labels, which get implied to the plots on the right; and the plots on the bottom have x-axis labels, which get implied to the plots in the middle and the top.

  1. The limitations section should include a discussion of the immediate and long-term effects of the medication.

RESPONSE: The immediate and long-term effects of dopaminergic medication are complex. Indeed, this relationship becomes increasingly important the longer patients take medication, with many eventually experiencing unexpected OFF states between scheduled doses as the number of years they take dopamine replacement medication increases. (This is/was the impetus for developing surgical implantation of deep brain stimulators for medication-refractory PD.) Accordingly, such a discussion is outside of the scope of the current manuscript and we aimed to control for these potential (known) influences via our inclusion criteria (i.e., mild-moderate disease, stable medication regimen). Additionally, our recent addition of a statistical test of UPDRS III scores indicated significant improvement after taking medication (see above).

Round 2

Reviewer 1 Report

Comments and Suggestions for Authors

I do not have further comments.